# Time and classical equations of motion from quantum entanglement via the Page and Wootters mechanism with generalized coherent states

Caterina Foti [1,2,3 ✉], Alessandro Coppo[1,2], Giulio Barni[1], Alessandro Cuccoli [1,2] & Paola Verrucchi [1,2,4]

We draw a picture of physical systems that allows us to recognize what "time" is by requiring consistency with the way that time enters the fundamental laws of Physics. Elements of the picture are two non-interacting and yet entangled quantum systems, one of which acting as a clock. The setting is based on the Page and Wootters mechanism, with tools from large-$N$ quantum approaches. Starting from an overall quantum description, we first take the classical limit of the clock only, and then of the clock and the evolving system altogether; we thus derive the Schrödinger equation in the first case, and the Hamilton equations of motion in the second. This work shows that there is not a "quantum time", possibly opposed to a "classical" one; there is only one time, and it is a manifestation of entanglement.

[1] Dipartimento di Fisica e Astronomia, Università di Firenze, Sesto Fiorentino, Italy. [2] INFN, Sezione di Firenze, Sesto Fiorentino, Italy. [3] QTF Centre of Excellence, Department of Applied Physics, School of Science, Aalto University, Aalto, Finland. [4] ISC-CNR, UOS Dipartimento di Fisica, Università di Firenze, Sesto Fiorentino, Italy. ✉email: catefoti@gmail.com

The notion of time is deeply rooted into our perception of reality, which is why, for centuries, time has entered Physics as a fundamental ingredient that is not to be questioned. Then, general relativity (GR) and quantum mechanics (QM) intervened in opposite directions: GR gave time the same status of position, while QM made time a parameter, external to the theory and not recognizable as an observable. While the introduction of "spacetime" in GR appears as an elegant intuition, fully consistent with classical physics, the fact that time cannot be treated as any other observable in QM is disturbing. As a consequence, discussions about the role of time in QM have been developed, leading to different proposals on how to overcome what seems a serious inconsistency of the theory. Reporting upon these discussions goes beyond the scope of this paper; therefore, in what follows we will only refer to the proposal that provides our starting point. This was introduced by D. N. Page and W. K. Wotters in 1983[1] to formalize the idea that the expression "at a certain time $t$" should be understood as "conditioned to a clock being in a state labeled by a certain value $t$." This proposal, to which we will refer as the "Page and Wootters (PaW) mechanism," is based on three assumptions: (i) the clock does not interact with the system to which it provides the parameter $t$, but (ii) it is entangled with it; moreover, (iii) clock and system together are in an eigenstate of the total Hamiltonian (with eigenvalue that can be set equal to zero, for the sake of simplicity and without loss of generality). The PaW mechanism has been extensively used, and its assumptions scrutinized, in the recent literature, both from the theoretical and the experimental viewpoint[2–16]. Discussing about the many questions and answers on the PaW mechanism is not the scope of this work; however, references throughout the paper should help the reader to navigate the relevant literature, and our viewpoint on the contribution that our results furnish to the overall discussion on the mechanism itself is described in the concluding section.

Most discussions about time in QM are aimed at understanding what is the status of time in the quantum description, as if there were no problem as far as one stays classical. However, if one believes that there do not exist quantum systems and classical ones, but rather that some quantum systems behave in a way that, under certain conditions, is efficiently described by the laws of classical physics, then there must be just one time. In other terms, the procedure used to identify what time is in QM must have a well-defined classical limit, fully consistent with classical physics and the way time enters the classical equations of motion (e.o.m.). We construct such a procedure, demonstrating that it consistently produces not only the Schrödinger equation for quantum systems but also the Hamilton e.o.m. for classical ones, with the parameter playing the role of time being the same in both cases. We tackle the quantum-to-classical crossover via the large-$N$ approach based on Generalized Coherent States (GCS) from refs. [17–21], where it is demonstrated that the theory describing a quantum system for which GCS can be constructed flows into a well-defined classical theory if few specific conditions upon its GCS hold in the $N \to \infty$ limit ($N$ quantifies the number of microscopic quantum components, sometimes referred to as the number of degrees of freedom or dynamical variables, in the literature). By "classical limit," we will hereafter mean the large-$N$ limit with the above conditions on GCS enforced.

In this work, we consider a quantum composite system made of two non-interacting and entangled objects, dubbed clock and evolving system, whose quantum state is expressed via a parametric representation with GCS for the clock (see Fig. 1); from this representation, a real parameter $\varphi$ emerges, with features that make it a good candidate for being recognized as time. We then take the classical limit for the clock only (see Fig. 2) and derive an equation for the physical states of the evolving system, which is

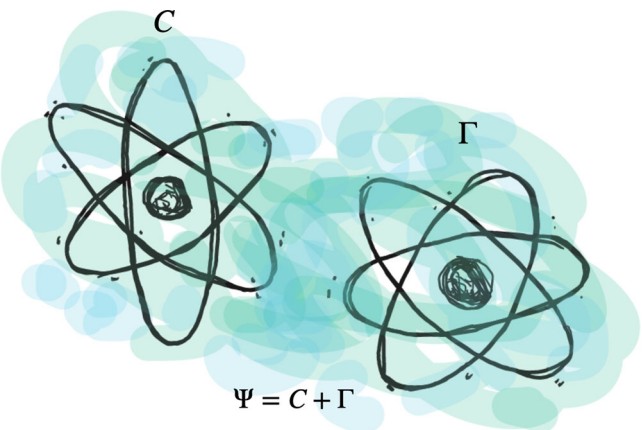

**Fig. 1 A quantum clock for a quantum system.** The clock $C$ and the evolving system $\Gamma$ make the isolated system $\Psi$, which is in the entangled state $|\Psi\rangle\rangle$.

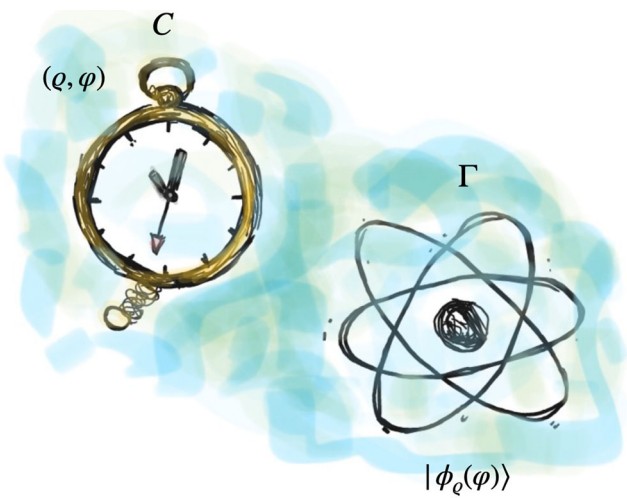

**Fig. 2 A classical clock for a quantum system.** The state of the classical clock $C$ is identified by the real variables $\varrho$ and $\varphi$, while $|\phi_\varrho(\varphi)\rangle$ is the state of the quantum system $\Gamma$ that parametrically depends on $(\varrho, \varphi)$.

the Schrödinger equation, once the above-mentioned parameter $\varphi$ is given the role of time. We also obtain an inequality that provides relevant clues for understanding the origin, nature, and meaning of the energy–time uncertainty relation. Finally, we introduce GCS for the evolving system, take its classical limit (see Fig. 3), and get to our most relevant result: the Hamilton e.o.m. of classical physics are derived, with the same parameter $\varphi$ as time.

## Results

**A quantum clock for a quantum system**. We consider a composite quantum system $\Psi = C + \Gamma$, with $C$ the clock and $\Gamma$ the evolving system; we assume that $\Psi$ is isolated, with Hamiltonian $\hat{H}$, and in a pure state $|\Psi\rangle\rangle$, which is entangled w.r.t. the partition $C$ and $\Gamma$; as in ref. [4], the double bracket indicates states in $\mathcal{H}_\Psi = \mathcal{H}_C \otimes \mathcal{H}_\Gamma$, with $\mathcal{H}_*$ the Hilbert space of $* = \Psi, C, \Gamma$.

Referring to the PaW mechanism, we enforce

$$\hat{H}|\Psi\rangle\rangle = 0 , \tag{1}$$

and take $C$ and $\Gamma$ non-interacting, i.e.

$$\hat{H} = \hat{H}_C \otimes \hat{\mathbb{1}}_\Gamma - \hat{\mathbb{1}}_C \otimes \hat{H}_\Gamma , \tag{2}$$

where the irrelevant minus sign in front of the term acting on $\Gamma$ is our choice for the sake of a lighter notation. As for the state $|\Psi\rangle\rangle$,

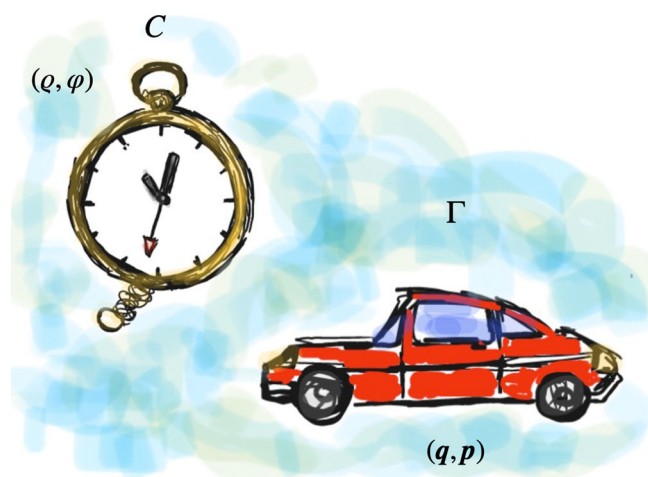

**Fig. 3 A classical clock for a classical system.** The state of the classical clock $C$ is identified by the real variables $\varrho$ and $\varphi$, while couples of canonically conjugated variables $(\boldsymbol{q}, \boldsymbol{p})$ are used to describe the state of the classical system $\Gamma$.

its most general expression is $|\Psi\rangle\rangle = \sum_{\gamma\xi} c_{\gamma\xi}|\xi\rangle \otimes |\gamma\rangle$, where $\{|\xi\rangle\}_C$ and $\{|\gamma\rangle\}_\Gamma$ are orthonormal bases of $\mathcal{H}_C$ and $\mathcal{H}_\Gamma$, respectively; the coefficients $c_{\gamma\xi} \in \mathbb{C}$ are such that $\sum_{\gamma\xi}|c_{\gamma\xi}|^2 = 1$ due to the normalization of $|\Psi\rangle\rangle$. Notice that if $|\Psi\rangle\rangle$ is entangled, there cannot exist orthonormal bases $\{|\xi\rangle\}_C$ and $\{|\gamma\rangle\}_\Gamma$ such that only one coefficient $c_{\gamma\xi}$ is different from zero.

In view of dealing with a parameter that must be continuous to represent time, we resort to a parametric representation (see "Methods") of $|\Psi\rangle\rangle$ with GCS for the clock, and write

$$|\Psi\rangle\rangle = \int_{\mathcal{M}_C} d\mu(\boldsymbol{\Omega})\chi(\boldsymbol{\Omega})|\boldsymbol{\Omega}\rangle \otimes |\phi(\boldsymbol{\Omega})\rangle, \tag{3}$$

where $|\boldsymbol{\Omega}\rangle$ are GCS defined via the group-theoretical construction[22,23] for the Lie group $\mathcal{G}_C$ associated with the algebra $\mathfrak{g}_C$ to which the Hamiltonian $\hat{H}_C$ belongs. The $M$-tuples $\boldsymbol{\Omega} = (\Omega_1, \Omega_2 \ldots \Omega_M)$, with $\Omega_m \in \mathbb{C} \,\forall m$, identify points on $\mathcal{M}_C$, which is a $2M$-dimensional manifold with a symplectic structure, and $M$ related to the dimension of $\mathfrak{g}_C$. The measure $d\mu(\boldsymbol{\Omega})$ is invariant w. r.t. the elements of $\mathcal{G}_C$ and ensures that GCS form a complete set upon $\mathcal{H}_C$, thus providing a resolution of the identity. The element $|\phi(\boldsymbol{\Omega})\rangle \in \mathcal{H}_\Gamma$ is normalized, and hence describes a physical state of $\Gamma$, parametrically dependent on $\boldsymbol{\Omega}$. Notice that the $\boldsymbol{\Omega}$-dependence of $|\phi(\boldsymbol{\Omega})\rangle$ survives iff $|\Psi\rangle\rangle$ is entangled. As for $\chi(\boldsymbol{\Omega})$, it is defined[24] (up to an arbitrary phase factor) through $\chi^2(\boldsymbol{\Omega}) = \sum_\gamma |\sum_\xi c_{\gamma\xi}\langle\boldsymbol{\Omega}|\xi\rangle|^2$, and can hence be taken real without loss of generality. The above function $\chi^2(\boldsymbol{\Omega})$ is a normalized probability distribution on $\mathcal{M}_C$ whose structure is strongly related to the entanglement property of $|\Psi\rangle\rangle$; in particular, if $|\Psi\rangle\rangle$ is entangled, $\chi^2(\boldsymbol{\Omega})$ is a superposition of different un-normalized distributions $|\sum_\xi c_{\gamma\xi}\langle\boldsymbol{\Omega}|\xi\rangle|^2$.

There is a certain degree of freedom in the group-theoretic construction of GCS (see, for instance, Tables I and II in ref. [25]), due to the possibility of choosing an arbitrary state $|G\rangle$ from which to start the construction, the so called reference state, and different sets of generators for $\mathfrak{g}_C$. For the non-semisimple algebra $\mathfrak{h}_4$ defining the harmonic-oscillator coherent states, for instance, it is customary to choose the set $\{\hat{a}^\dagger, \hat{a}, \hat{a}^\dagger\hat{a}, \hat{\mathbb{1}}\}$. When the semisimple Lie algebras $\mathfrak{su}(2)$ or $\mathfrak{su}(1,1)$, defining the spin or pseudo-spin coherent states, respectively, are considered, the standard choice is the set $\{\hat{S}^-, \hat{S}^+, \hat{S}_z\}$ in the first case and $\{\hat{K}^-, \hat{K}^+, \hat{K}_0\}$ in the second one, being $\hat{S}_z \setminus \hat{K}_0$ diagonal operators

and $\hat{S}^-, \hat{S}^+ \setminus \hat{K}^-, \hat{K}^+$ raising ones. In fact, in the case of semisimple algebras different choices of the generators are generally embedded in a different choice of the Cartan decomposition (and hence of the so called Cartan basis). To this respect, we remind that the Cartan decomposition classifies generators into diagonal, $\{\hat{D}_\delta\}$, and raising operators, $\{\hat{R}_m, \hat{R}_{-m}\}$, according to $[\hat{D}_\delta, \hat{D}_\theta] = 0, [\hat{D}_\delta, \hat{R}_m] = d_{\delta m}\hat{R}_m, [\hat{R}_m, \hat{R}_{-m}] = \sum_\delta d_{\delta m}\hat{D}_\delta$, and $[\hat{R}_m, \hat{R}_{m'}] = c_{mm'}\hat{R}_{m+m'}$, where the coefficients $\{d_{\delta m}\}, \{c_{mm'}\}$ are the so called structure constants. Referring again to the familiar cases of $\mathfrak{su}(2)$ and $\mathfrak{su}(1,1)$, it is $M = 1$, with $\hat{R}_1 = \hat{S}^-$ and $\hat{K}^-$. If spin squeezing is also considered, it is $M = 2$, with $\hat{R}_2 = (\hat{S}^-)^2$. Let us finally remind that the harmonic-oscillator algebra $\mathfrak{h}_4$ is not semisimple, and a Cartan decomposition cannot hence be defined for it. Nevertheless, the usual choice of generators corresponds to an analogous decomposition, with $(\hat{a}^\dagger\hat{a}, \hat{\mathbb{1}})$ diagonal and $(\hat{a}^\dagger, \hat{a})$ creation/annihilation operators. Given the relevance of the harmonic-oscillator coherent states, we have specifically addressed their case at the end of the "Methods" section.

We choose the Cartan basis so that $\hat{H}_C$ depends linearly on one of its diagonal operators only, say $\hat{H}_C = \varsigma\hat{D}_1 + k\hat{\mathbb{1}}_C$, where $k$ is a real arbitrary constant and $\varsigma^2 = \pm 1$ such that $\epsilon := \varsigma d_{1\ell}$ is real and positive for some $\ell$, which ensures $\hat{H}_C$ is Hermitian. Notice that $\varsigma^2 = +1$ or $-1$ depending on whether $\mathcal{M}_C$ is compact or non-compact, respectively; consequently, $\varsigma$ is either 1 or $i$, and the functions having argument proportional to $\varsigma$ have a different nature, trigonometric or hyperbolic, respectively, throughout the manuscript (see Sec.3C.1.a of ref. [25] for more details). As for the parameter $\epsilon$, due to the definition of $\hat{H}_C$ it embodies both $\hbar$ and the energy scale of the clock. For the sake of a lighter notation, we also normalize the raising and diagonal operators so that $\varsigma^2\sum_\delta d_{\delta\ell}^2 \to 2$. As for the reference $|G\rangle$, we set it as the minimal weight state, $\hat{R}_m|G\rangle = 0 \,\forall m$, which is easily seen to be an eigenstate of the diagonal operators, $\hat{D}_\delta|G\rangle = g_\delta|G\rangle$. In particular, hence, it is $\hat{H}_C|G\rangle = \epsilon_0|G\rangle$, with $\epsilon_0 := \varsigma g_1 + k$, and we will hereafter take $k$ so that $\epsilon_0 = 0$.

Once the Cartan basis and the reference state are chosen, GCS are generated via

$$|\boldsymbol{\Omega}\rangle = e^{\boldsymbol{\Omega}\cdot\hat{\mathbf{R}}^\dagger - \boldsymbol{\Omega}^*\cdot\hat{\mathbf{R}}}|G\rangle, \tag{4}$$

where $\hat{\mathbf{R}} := (\hat{R}_1, \hat{R}_2 \ldots \hat{R}_M)$; notice that the index $m$ runs from 1 to $M$ both in $\Omega_m$ and in $\hat{R}_m$, by definition. GCS as from Eq. (4) are normalized and non-orthogonal, and expectation values of operators upon them, $\langle\boldsymbol{\Omega}|\hat{O}|\boldsymbol{\Omega}\rangle$, are often dubbed symbols, indicated by $O(\boldsymbol{\Omega})$. For more technical details on this section, we refer the reader to the "Methods" section.

We consider the set of GCS defined by $\boldsymbol{\Omega}_\ell = (0, 0, \ldots, \Omega_\ell, \ldots 0)$, with $\ell$ chosen at will among those for which $\epsilon$ is real and positive, and $\chi(\boldsymbol{\Omega}_\ell) \neq 0$. Given that $\Omega_\ell \in \mathbb{C}$, we will hereafter use

$$\lambda := \Omega_\ell = \varrho e^{-i\varphi}, \tag{5}$$

with $\varrho \in [0, \infty)$ and $\varphi \in (-\infty, \infty)$. Using the BCH formulas proper to $\mathfrak{g}_C$, and the definition (4), it can be easily shown that

$$|\lambda\rangle := |\boldsymbol{\Omega}_\ell\rangle = N_\varrho e^{\Lambda\hat{R}_\ell^\dagger}|G\rangle, \tag{6}$$

with $\Lambda = |\tan(\varsigma\varrho)|e^{-i\varphi}$ and $N_\varrho$ a normalization factor that does not depend on $\varphi$. Furthermore, from the Cartan commutation rule $[\hat{D}_\delta, \hat{R}_\ell] = d_{\delta\ell}\hat{R}_\ell$ it follows $[\hat{H}_C, e^{\Lambda^*\hat{R}_\ell}] = \epsilon\Lambda^*\hat{R}_\ell e^{\Lambda^*\hat{R}_\ell}$, leading to

$$\langle\lambda|\hat{H}_C|\boldsymbol{\Omega}\rangle = \langle G|N_\varrho e^{\Lambda^*\hat{R}_\ell}\hat{H}_C|\boldsymbol{\Omega}\rangle = i\epsilon\frac{d}{d\varphi}\langle\lambda|\boldsymbol{\Omega}\rangle. \tag{7}$$

Once defined the partial inner product $\langle\cdot|\cdot\rangle\rangle : \mathcal{H}_C \otimes \mathcal{H}_\Gamma \to \mathcal{H}_\Gamma$ such that $\langle\zeta|[\langle\xi|\Psi\rangle\rangle] = (\langle\zeta| \otimes \langle\xi|)|\Psi\rangle\rangle, \forall\xi \in \mathcal{H}_C$, and $\forall\zeta \in \mathcal{H}_\Gamma$,

we project the constraint (1) in the form

$$\langle\lambda|\hat{H}|\Psi\rangle = 0\,, \tag{8}$$

with $\hat{H}$ and $|\Psi\rangle$ as in Eqs. (2) and (3), and find, by virtue of the result (7),

$$i\epsilon\frac{d}{d\varphi}|\Phi_\varrho(\varphi)\rangle = \hat{H}_\Gamma|\Phi_\varrho(\varphi)\rangle\,, \tag{9}$$

where

$$|\Phi_\varrho(\varphi)\rangle := \langle\lambda|\Psi\rangle = \int_{\mathcal{M}_C} d\mu(\mathbf{\Omega})\chi(\mathbf{\Omega})\langle\lambda|\mathbf{\Omega}\rangle|\phi(\mathbf{\Omega})\rangle \tag{10}$$

is an un-normalized element of $\mathcal{H}_\Gamma$, and we have introduced a notation that highlights the different meaning that the dependence on $\varrho$ will have in what follows, w.r.t. that on $\varphi$. Reminding that $\epsilon$ is real and positive, Eq. (9) has the same form of the Shrödinger equation, with the real parameter

$$\frac{\hbar}{\epsilon}\varphi \tag{11}$$

playing the role of time, as found resorting to other parametric representations[1,4,14,26]. However, Eq. (9) is not the Schrödinger equation, as $|\Phi_\varrho(\varphi)\rangle$ is not normalized. This is most often considered an amendable fault, as from Eq. (9) it follows $\frac{d}{d\varphi}\langle\Phi_\varrho(\varphi)|\Phi_\varrho(\varphi)\rangle = 0$ meaning that, should $|\Phi_\varrho(\varphi)\rangle$ have a non-vanishing and finite norm, Eq. (9) would also hold for its normalized sibling. Before considering this point, let us collect some more clues on the meaning of $\varrho$ and $\varphi$.

Getting back to the operator $\hat{R}_\ell$ introduced at the beginning of this section, one can define[27,28] the so called "phase-operator" $\hat{\phi}$, via

$$\hat{R}_\ell = (\hat{R}_\ell\hat{R}_\ell^\dagger)^{1/2}e^{-i\hat{\phi}}\,. \tag{12}$$

From the commutation rules between elements of the Cartan basis, reminding that $\hat{H}_C = \varsigma\hat{D}_1 + k\hat{\mathbb{I}}_C$ and $\epsilon = \varsigma d_{1\ell} \in \mathbb{R}^+$, it follows

$$[\hat{H}_C, \sin\hat{\phi}] = i\epsilon\cos\hat{\phi}\,, \tag{13}$$

and hence (see, for instance, ref. [29])

$$\Delta\hat{H}_C\Delta\sin\hat{\phi} \geq \left|\frac{\epsilon}{2}\langle\cos\hat{\phi}\rangle\right|\,, \tag{14}$$

with $\Delta\hat{B} := (\langle\hat{B}^2\rangle - \langle\hat{B}\rangle^2)^{1/2}$ for any Hermitian operator $\hat{B}$. Noticing that Eqs. (1) and (2) imply a relation between $\hat{H}_C$ and the energy of the system, while Eqs. (5) and (12) relate $\hat{\phi}$ with $\varphi$, one might say that the inequality (14) is the ancestor of the time–energy uncertainty relation for $\Gamma$, after setting $\varphi \ll 1$ and the parameter (11) precisely as time, a statement that is made clear in the next section. We also remind that the choice of $\ell$ is arbitrary, as long as it ensures $\epsilon$ real and positive, and $\chi(\mathbf{\Omega}_\ell) \neq 0$. So, the freedom left by this mild request allows one to further ask that time can be read by observing the clock. This means that there must exist Hermitian linear combinations of the operators $\hat{R}_\ell$ and $\hat{R}_\ell^\dagger$ that are experimentally accessible observables for $C$, and such that the results of their measurements carry information upon the coordinate $\varphi$.

Summarizing, we have so far collected results that point to $\hbar\varphi/\epsilon$ as "the time" for the evolving system, but the overall picture is not that provided by QM, where the quantum character of the clock is totally absent; this is the reason why we take our next step.

**A classical clock for a quantum system**. We now assume that the quantum theory describing $C$ satisfies the conditions ensuring it flows into a well-defined classical theory when the clock becomes macroscopic, according to the large-$N$ quantum approach based on GCS, as briefly described in the "Introduction" section. In particular, we use that GCS are the only quantum states that survive the quantum-to-classical crossover, insofar doing becoming orthogonal

$$\lim_{N\to\infty}\langle\mathbf{\Omega}|\mathbf{\Omega}'\rangle \to \delta(\mathbf{\Omega} - \mathbf{\Omega}')\,, \tag{15}$$

and defining the classical states identified by the corresponding points $\mathbf{\Omega}$ on the classical phase-space $\mathcal{M}$. As for the observables, the only ones that stay meaningful throughout the crossover must obey

$$\lim_{N\to\infty}\frac{\langle\mathbf{\Omega}|\hat{A}|\mathbf{\Omega}'\rangle}{\langle\mathbf{\Omega}|\mathbf{\Omega}'\rangle} < \infty\,, \tag{16}$$

so as to transform into well-defined functions on the classical phase-space. Using Eq. (15), one can easily show that $\langle\Phi_\varrho(\varphi)|\Phi_\varrho(\varphi)\rangle \to \chi^2(\lambda)$ in the classical limit for the clock; moreover, it is $\chi^2(\lambda) \equiv \chi^2(\varrho)$ due to Eq. (9). Therefore, reminding that $\chi^2(\varrho)$ is a normalized probability distribution, any $\varrho$ for which $\chi^2(\varrho) \neq 0$ defines a physical state

$$|\phi_\varrho(\varphi)\rangle := \frac{|\Phi_\varrho(\varphi)\rangle}{\sqrt{\chi^2(\varrho)}}\,, \tag{17}$$

whose dependence on $\varphi$ is ruled by

$$i\epsilon\frac{d}{d\varphi}|\phi_\varrho(\varphi)\rangle = \hat{H}_\Gamma|\phi_\varrho(\varphi)\rangle\,, \tag{18}$$

which is the Schrödinger equation with $t = \hbar\varphi/\epsilon$. In fact, the above result is a derivation of the Schrödinger equation akin to that suggested in the original work by Page and Wootters[1], with state normalization ensured by construction, for a classical clock. We notice, though, that as a byproduct of having specifically addressed the normalization issue, the state (17) has a further dependence on the real parameter $\varrho$. In order to understand its meaning as far as the evolving system is concerned, we get back to the constraint (1) and its projection upon a GCS $|\lambda\rangle$ of the clock, Eq. (8), with $|\Psi\rangle$ as in Eq. (3). Considering that $\langle\lambda|\mathbf{\Omega}\rangle$ is finite for finite $N$, we write

$$0 = \langle\lambda|\hat{H}|\Psi\rangle$$
$$= \int_{\mathcal{M}} d\mu(\mathbf{\Omega})\chi(\mathbf{\Omega})\langle\lambda|\mathbf{\Omega}\rangle\left(\frac{\langle\lambda|\hat{H}_C|\mathbf{\Omega}\rangle}{\langle\lambda|\mathbf{\Omega}\rangle} - \hat{H}_\Gamma\right)|\phi(\mathbf{\Omega})\rangle \tag{19}$$

that becomes, in the classical limit for $C$ where Eqs. (15) and (16) hold and for any $\varrho$ such that $\chi^2(\varrho) \neq 0$,

$$\hat{H}_\Gamma|\phi_\varrho(\varphi)\rangle = E_\Gamma(\varrho)|\phi_\varrho(\varphi)\rangle\,, \tag{20}$$

with

$$E_\Gamma(\varrho) = \langle\lambda|\hat{H}_C|\lambda\rangle\,; \tag{21}$$

the r.h.s. of the above equation, which is the symbol of $\hat{H}_C$ on $|\lambda\rangle$, can be calculated and reads (see in "Methods")

$$H_C(\varrho) := \langle\lambda|\hat{H}_C|\lambda\rangle = \frac{\epsilon}{2}b^2(\cos(2\varsigma\varrho) - 1)\,, \tag{22}$$

with $\varsigma^2b^2 = \sum_\delta g_\delta d_{\delta\ell}$. It is relevant that Eq. (22) follows from algebraic properties, and therefore holds in general, regardless of the details of the theory that describes the clock. Furthermore, $H_C(\varrho)$ does not depend on $\varphi$, which justifies the use of the notation $E_\Gamma(\varrho)$ in Eq. (21) and allows one to consistently relate Eq. (20) with the stationary Schrödinger equation for $\Gamma$, with $\varrho$ the parameter that sets its energy.

Let us now consider what happens when making measurements on the clock. We know that GCS are the only quantum states that survive the quantum-to-classical crossover according to $|\Omega\rangle \rightarrow \Omega$, as described above and thoroughly discussed in the literature[30–34]. This means that performing a quantum measurement upon a system whose behavior can be effectively described as if it were classical is tantamount to select one GCS $|\Omega\rangle$ to be the ancestor of the observed classical state or, which is the same, say that the combined effect of a measurement and the classical limit is to make $\chi^2(\Omega)$ become a Dirac-$\delta$ around the point $\Omega$ on $\mathcal{M}_C$ that identifies the observed classical state. Let us now take such state to be one of the GCS $|\lambda\rangle$, consistently with the task of making measurements of observables that characterize it as a clock, such as $\hat{H}_C$ or $\sin\hat{\phi}$ in Eq. (13). When taking the classical limit of the clock, it can be demonstrated[27,28] that

$$\langle\lambda|\sin\hat{\phi}|\lambda\rangle \rightarrow \sin\varphi \ , \ \ \langle\lambda|\cos\hat{\phi}|\lambda\rangle \rightarrow \cos\varphi\,; \quad (23)$$

this result, together with the definition $\Delta E_\Gamma(\varrho) := \Delta H_C(\varrho)$ (that follows from Eqs. (20) and (21)) and a small-$\varphi$ approximation, provides

$$\Delta E_\Gamma(\varrho)\Delta\varphi \geq \frac{\epsilon}{2}\,, \quad (24)$$

where $\Delta\varphi$ is related to the mean square fluctuation of the operator $\hat{\varphi}$ for the clock via the classical limit of the clock itself. If the parameter $\hbar\varphi/\epsilon$ is identified with time, the above inequality has the form of an energy–time uncertainty relation for $\Gamma$. However, $\Delta\varphi$ is not a time interval, and the way the result (24) is obtained suggests an altogether different interpretation, w.r.t. the usual one, of the elements entering the energy–time uncertainty relation. We will further comment upon this result in the concluding section.

Collecting all the clues so far obtained, we conclude this section maintaining that the parameter (11) is what we call "time" in QM, a statement that we express as

$$t^{\mathrm{QM}} = \frac{\hbar}{\epsilon}\varphi\,, \quad (25)$$

where the apex QM indicates that this is the parameter that enters the quantum description of evolving systems.

This is not the end of the story, though, because it is now necessary to demonstrate that when the system $\Gamma$ undergoes the quantum-to-classical crossover, the above results lead to the Hamilton e.o.m., with the parameter $\hbar\varphi/\epsilon$ still playing the role of time. To this purpose, in the next section we take the classical limit also for the evolving system, thus moving into a completely classical setting.

**A classical clock for a classical system**. Let us now consider what happens when the system $\Gamma$ becomes macroscopic in a way that makes its behavior amenable to the laws of classical physics. As in the previous section, the problem is tackled in terms of GCS in the large-$N$ limit. Therefore, besides the GCS for the clock $\{|\Omega\rangle\}$, here we also use the GCS for the system, i.e., those relative to the Lie algebra $\mathfrak{g}_\Gamma$ proper to the quantum theory that describes $\Gamma$. These will be indicated by $\{|\gamma\rangle\}$, where $\gamma = (\gamma_1, \gamma_2, \ldots \gamma_J)$ with $\gamma_j \in \mathbb{C} \ \forall j$, and $J$ related to the dimension of $\mathfrak{g}_\Gamma$. Each $|\gamma\rangle$ univocally identifies one point on the manifold $\mathcal{M}_\Gamma$, whose (real) dimension is $2J$.

Using the resolution of the identity upon $\mathcal{H}_C$ and $\mathcal{H}_\Gamma$ in terms of the GCS $\{|\Omega\rangle\}$ and $\{|\gamma\rangle\}$, respectively, we write the state $|\Psi\rangle$ of the overall system as

$$|\Psi\rangle = \int_{\mathcal{M}_C} d\mu(\Omega) \int_{\mathcal{M}_\Gamma} d\mu(\gamma)\beta(\Omega, \gamma)|\Omega\rangle \otimes |\gamma\rangle\,, \quad (26)$$

where

$$\beta(\Omega, \gamma) := (\langle\Omega| \otimes \langle\gamma|)|\Psi\rangle = \chi(\Omega)\langle\gamma|\phi(\Omega)\rangle \quad (27)$$

is a function on $\mathcal{M}_C \times \mathcal{M}_\Gamma$ whose square modulus, $\chi^2(\Omega)|\langle\gamma|\phi(\Omega)\rangle|^2$, is the conditional probability for $\Gamma$ to be in the state $|\gamma\rangle$ when $C$ is in the state $|\Omega\rangle$, given that the global system $\Psi$ is in the pure state $|\Psi\rangle$. In other terms, $\beta(\Omega, \gamma)$ is different from zero only on those pairs $(\Omega, \gamma) \in \mathcal{M}_C \times \mathcal{M}_\Gamma$ that define states $|\Omega\rangle \otimes |\gamma\rangle \in \mathcal{H}_C \otimes \mathcal{H}_\Gamma$, which are present in the decomposition of $|\Psi\rangle$ in terms of GCS, Eq. (26).

Projecting the constraint (1) upon one specific state $|\overline{\Omega}\rangle \otimes |\overline{\gamma}\rangle$, we write

$$\begin{aligned} 0 &= \langle\overline{\Omega}| \otimes \langle\overline{\gamma}|\hat{H}|\Psi\rangle \\ &= \int_{\mathcal{M}_C} d\mu(\Omega) \int_{\mathcal{M}_\Gamma} d\mu(\gamma)\beta(\Omega, \gamma)\langle\overline{\Omega}|\Omega\rangle\langle\overline{\gamma}|\gamma\rangle \\ &\quad \times \left[\frac{\langle\overline{\Omega}|\hat{H}_C|\Omega\rangle}{\langle\overline{\Omega}|\Omega\rangle} - \frac{\langle\overline{\gamma}|\hat{H}_\Gamma|\gamma\rangle}{\langle\overline{\gamma}|\gamma\rangle}\right] \end{aligned} \quad (28)$$

that becomes, in the classical limit for $C$ and $\Gamma$, i.e., assuming Eqs. (15) and (16) hold not only for the GCS and the Hamiltonian of the clock but also for those of the system,

$$H_C(\Omega) = H_\Gamma(\gamma) \quad (29)$$

for $(\Omega, \gamma)$ such that $\beta(\Omega, \gamma) \neq 0$, meaning that the configurations $(\Omega, \gamma)$ into which the original quantum state $|\Psi\rangle$ can flow when clock and system behave according to the rules of classical physics, must obey Eq. (29). In particular, if one considers the configurations among those for which $\beta(\Omega, \gamma) \neq 0$ that have $\Omega = (0, 0, \ldots \Omega_\ell, \ldots 0)$, corresponding to the GCS $|\lambda\rangle$ introduced above and identified by the complex variable $\lambda = \varrho e^{-i\varphi}$, these will belong to a submanifold $(\mathcal{U}_C \subset \mathbb{C}) \times (\mathcal{U}_\Gamma \subset \mathcal{M}_\Gamma)$ such that a map $F : \mathcal{U}_C \rightarrow \mathcal{U}_\Gamma$ exists, defined by

$$\lambda \in \mathcal{U}_C \xrightarrow{F} \mathbf{u} \in \mathcal{U}_\Gamma : H_\Gamma(\mathbf{u} = F(\lambda)) = H_C(\varrho)\,. \quad (30)$$

As the explicit form of $F$ is arbitrary, we fix it as follows. We consider that $\mathcal{M}_\Gamma$ has a symplectic structure, which means that it exists a Darboux chart

$$\begin{cases} \mathcal{D} : \gamma \in \mathcal{M}_\Gamma \ \rightarrow (\mathbf{q}, \mathbf{p}) := ((q_1, p_1), (q_2, p_2)\ldots(q_J, p_J)) \in \mathbb{R}^{2J}\,, \\ \text{such that } \{q_i, p_j\}_\Gamma = \hbar^{-1}\delta_{ij} \text{ with } \hbar = \text{const.}\,, \\ \text{where } \{\cdot, \cdot\}_\Gamma \text{ are Poisson brackets on } \mathcal{M}_\Gamma\,, \end{cases} \quad (31)$$

that relates the parametrization of GCS via $J$-dimensional complex vectors $\{\gamma\}$ with that obtained via $J$ pairs of real, canonically conjugated, variables $(q_j, p_j)$. For these pairs, referring to ref. [25], we choose

$$q_j - i\varsigma^2 p_j = v_j\sqrt{2}b\varsigma\sin(\varsigma\varrho)e^{-i\varphi} \quad (32)$$

with $\overrightarrow{v} \in \mathbb{R}^J$ constant unit vector, i.e., $\sum_j v_j^2 = 1$. As far as condition (30) is fulfilled, other choices are possible, without affecting the overall scheme and the subsequent results. Once $F$ is given, the so called "pullback-by-$F$" map, sometimes indicated by $F^*$, is also defined, according to $F^* : \omega_\Gamma^{(\kappa)} \longrightarrow \omega_C^{(\kappa)}$, where $\omega_{\Gamma(C)}^{(\kappa)}$ are $\kappa$-forms on $\mathcal{U}_{\Gamma(C)}$. In particular, for $\kappa = 0$, i.e., when considering functions, it is $(F^*f_\Gamma)(\lambda) = f_C(\lambda)$, with $F^*f_\Gamma = f_\Gamma(\mathbf{u} = F(\lambda))$. Applying $F^*$ on the symplectic 2-form defining the standard Poisson brackets in (31), we obtain the Poisson brackets on $\mathcal{M}_C$, that read (see in "Methods")

$$\{f_C, g_C\}_C = \frac{1}{\hbar b^2\varsigma\sin(2\varsigma\varrho)}\left(\frac{\partial f_C}{\partial\varrho}\frac{\partial g_C}{\partial\varphi} - \frac{\partial g_C}{\partial\varrho}\frac{\partial f_C}{\partial\varphi}\right) \quad (33)$$

$\forall f_C, g_C$ generic functions on $\mathcal{M}_C$. On the other hand, $q_j$ and $p_j$ are by all means functions on $\mathcal{M}_C$, as seen in Eq. (32); therefore, using Eq. (33) with $g_C = H_C(\varrho)$ from Eq. (22) we evaluate $\{q_j, H_C\}_C$ and $\{p_j, H_C\}_C$, and find (see in "Methods") $\{q_j, H_C\}_C = \frac{\epsilon}{\mathfrak{h}} \frac{d\,q_j}{d\,\varphi}$, and $\{p_j, H_C\}_C = \frac{\epsilon}{\mathfrak{h}} \frac{d\,p_j}{d\,\varphi}$. Finally, using $\{f_C(\lambda), g_C(\lambda)\}_C = \{f_\Gamma(\mathbf{u}), g_\Gamma(\mathbf{u})\}_\Gamma$, we obtain

$$\begin{cases} \{q_j, H_\Gamma\}_\Gamma = \frac{\epsilon}{\mathfrak{h}} \frac{d\,q_j}{d\,\varphi} \\ \{p_j, H_\Gamma\}_\Gamma = \frac{\epsilon}{\mathfrak{h}} \frac{d\,p_j}{d\,\varphi} \end{cases} \tag{34}$$

i.e., the Hamilton e.o.m. ruling the dynamics of classical systems, once time is recognized as the parameter

$$t^{\mathrm{CL}} = \frac{\mathfrak{h}}{\epsilon} \varphi\,, \tag{35}$$

where the apex CL indicates that this is the parameter that enters the classical description of evolving systems. Getting back to Eq. (25) and setting the arbitrary constant $\mathfrak{h}$ in the Poisson brackets of the Darboux chart (31) equal to $\hbar$, consistently with the fact that $\epsilon$ is defined as $\hbar$ times a quantity that sets the energy scale of the clock, we finally obtain

$$t^{\mathrm{QM}} = t^{\mathrm{CL}} = \frac{\hbar}{\epsilon} \varphi\,, \tag{36}$$

This last equation, together with the derivation in one same framework of both the quantum-mechanical Schrödinger equation (18) and the classical Hamilton e.o.m. (34), represents the main result of this work, which is discussed in the next and last section.

## Discussion

In the past decades, we have learnt that when quantum macroscopic systems can be effectively studied as if they were classical (which is what should be meant by "classical"), their geometrical properties follow from the algebraic structure of the quantum theory originally describing them (see, for instance, the way a specific phase-space emerges as the symplectic manifold involved in the GCS construction for one assigned quantum Lie algebra). This is by itself quite a breakthrough, as it allows to establish a dialog between classical and quantum physics without resorting to disjointed interventions such as quantization or, in the opposite direction, non-unitary state reduction.

When considering more than one system, things become ever more interesting. In fact, when a quantum system interacts with a classical environment (be that a magnetic field, or a thermal bath, or some macroscopic environment), the pure states of the former acquire a parametric dependence that testifies the existence of the latter and gives rise to geometrical effects such as the quantum Berry phase[24,35–37]. Awe comes, though, as these effects emerge even without interaction, as far as the systems are entangled and some physical constraint is enforced, such as Eq. (1) in the PaW mechanism. Indeed, this is how states of a quantum system come to depend on time according to the Schrödinger equation, as also shown in this work. In such setting, coordinates of points in manifolds and elements of Hilbert spaces (e.g., $\varrho$, $\varphi$ and $|\phi_\varrho(\varphi)\rangle$ in this work) relate to each other via rules, such as the Schrödinger equation or the time–energy uncertainty relation, whose generality is that of the physical principles. To this respect, we like to comment upon two of our results: First, we notice that the energy of the system $\Gamma$, i.e., $E_\Gamma(\varrho)$ in Eq. (20), does not depend on time, i.e., on $\varphi$, consistently with the fact that the Hamiltonian of an isolated system cannot depend on time. Then we underline that the inequality (24) does not follow from the non-commutativity between $\hat{H}_\Gamma$ and some other operator acting on $\mathcal{H}_\Gamma$: it is rather an indirect consequence of the inequality (13), which regards

operators acting on the clock, plus the constraint (1) and the possibility, given by the use of GCS, of describing the clock as a classical object without wiping out the effects of one of the most relevant quantum feature, namely, entanglement. It is also worth reminding that $\Delta\varphi$ in the inequality (24) does not emerge as a time interval, which suggests that such inequality, despite having the same form of the customary energy–time uncertainty relation, is of a somewhat different nature and origin. This relevant point will be further investigated in future work.

Overall, it is indeed remarkable that effects of a genuinely quantum feature such as entanglement survive in a completely classical setting, there continuing to cause, via the correlation established between clock and system, the emergence of such a fundamental ingredient of our everyday life as time, which is what we have here demonstrated by deriving the Hamilton e.o.m (34). Indeed, the fact that fundamental laws of classical Physics be derived within a completely quantum framework with the PaW-mechanism assumptions enforced substantiates the mechanism itself. Moreover, our results in the fully classical setting unravel another tangle of classical physics, namely, the relation between phase-space and spacetime. This relation emerges from the fact that, when the global system is in the pure state $|\Psi\rangle\rangle$, the only configurations that survive its classical limit are those identified by points $(\Omega, \gamma) \in \mathcal{M}_C \times \mathcal{M}_\Gamma$ where the probability $|\beta(\Omega, \gamma)|^2$ is different from zero. Therefore, while the phase-space of $\Gamma$ is the $2J$ dimensional symplectic manifold $\mathcal{M}_\Gamma$ defined by the GCS $|\gamma\rangle$, its spacetime is the $(J + 1)$-dimensional real hypersurface defined by Eqs. (29) and (32), whose points are identified by the coordinates $(\hbar\varphi/\epsilon; \mathbf{q})$, with $\varphi = -\arg\lambda \in \mathbb{R}$ from Eq. (5) and $\mathbf{q} = \mathbf{q}(\gamma) \in \mathbb{R}^J$ from the Darboux chart (31), such that $\beta(\overline{\varrho}, \varphi; \mathbf{q}, \overline{\mathbf{p}})$ is different from zero for some $\overline{\varrho}$ (i.e., energy of the clock) and $\overline{\mathbf{p}}$ (i.e., momentum of the system). For instance, if $\Gamma$ is a particle in a 3-dimensional space, it is $\mathbf{q} = (x, y, z)$, $J = 3$, and one finds the $J + 1 = 4$-dimensional spacetime. Notice that, if $C$ and $\Gamma$ were not entangled, i.e., $|\Psi\rangle\rangle = |C\rangle \otimes |\Gamma\rangle$, it would be $\beta(\overline{\varrho}, \varphi; \mathbf{q}, \overline{\mathbf{p}}) = \chi(\overline{\varrho})\langle\langle(\mathbf{q}, \overline{\mathbf{p}})|\Gamma\rangle$, with no relation between instants of time $\hbar\varphi/\epsilon$ and position in space $\mathbf{q}$. In other terms, as emerged in different contexts (see, for instance, ref. [38]) not only quantum entanglement is what makes physical systems to evolve but it also provides their spacetime with a causal structure.

Despite effects of entanglement without interaction being already phenomenal, we think that taking possible interactions into account will lead to substantial developments of this work. One might first consider adding a quantum environment with which $\Gamma$ starts interacting while being already entangled with the clock. This should describe the dynamics of the density operator of $\Gamma$ and show how, and under what conditions, the Liouville–VonNeumann equation emerges, with clues about the non-unitary evolution of non-isolated systems. The presence of multiple clocks, possibly interacting among themselves, also seems an intriguing enrichment, particularly in view of some recent works by other authors[11,16]. However, the most compelling follow-up of this work, in our opinion, is that of relating the picture it proposes with that provided by relativity. In fact, we expect relativistic quantum mechanics and quantum field theory to find their place in the hybrid setting described in the section "A classical clock for a quantum system," where studying how the expectation values of operators on $\mathcal{H}_\Gamma$ get to depend on $(\varrho, \varphi)$ via the parametric dependence of the states $|\phi_\varrho(\varphi)\rangle$, might help understanding some unclear aspects of the way special relativity encounters quantum mechanics. Moreover, having connected the classical formalism that set the scene for general relativity and gravity with a full quantum description, we think we have ideal tools for breaking through some of the obstacles that make quantum gravity so difficult to process. In particular, we believe that studying the probability distribution $|\beta(\lambda; \mathbf{q}, \mathbf{p})|^2$ in relation

to the original Lie algebras $\mathfrak{g}_C$ and $\mathfrak{g}_\Gamma$ and/or the specific form of the quantum Hamiltonian $H$ may provide a link between the geodesic principle and the Schrödinger equation; furthermore, taking into account a possible interaction between evolving system and clock, as suggested in ref. [11], or between different clocks, as in ref. [16], might explain spacetime deformation, and hence gravity, from a quantum viewpoint. Work in this direction is in progress, particularly referring to the case of Schwartzschild black holes and Hawking radiation.

## Methods

**Generalized Coherent States.** GCS are an extension of the field coherent states firstly introduced by R. Glauber in 1963[39]. The group-theoretic construction was derived ten years later by A. Perelomov[22] and R. Gilmore[23], independently. GCS are normalized elements of Hilbert spaces which are in one-to-one correspondence with the points of a smooth manifold, that has all the properties requested to a classical phase-space. In the following, we briefly introduce GCS according to the procedure described by Gilmore and co-workers in ref. [25].

In order to construct GCS, three inputs are necessary:

(i1) a Lie algebra $\mathfrak{g}$, or the related Lie group $\mathcal{G}$,

(i2) a Hilbert space $\mathcal{H}$, which is the carrier space of an irreducible representation of $\mathfrak{g}$, and

(i3) a normalized element $|G\rangle$ of $\mathcal{H}$.

Referring to a specific system for which GCS are to be constructed, the inputs are as follows: $\mathcal{H}$ is the Hilbert space of the system; $\mathfrak{g}$ is the Lie algebra whose representation via operators on $\mathcal{H}$ contains the Hamiltonians of the system, meaning that the representation of the related Lie group $\mathcal{G}$ contains all its propagators, which is why $\mathcal{G}$ is often dubbed dynamical group. The normalized element $|G\rangle$ of $\mathcal{H}$ is a physically accessible state of the system, usually called reference state. For the sake of clarity, we will hereafter identify $\mathfrak{g}$ and $\mathcal{G}$ with their respective representations on $\mathcal{H}$. Once the inputs are given, the procedure returns three outputs:

(o1) the subgroup $\mathcal{F} \subset \mathcal{G}$ whose elements leave $|G\rangle$ unchanged apart from an irrelevant overall phase, and the associated coset $\mathcal{G}/\mathcal{F}$, such that every $\hat{\mathbf{g}} \in \mathcal{G}$ can be written as a unique decomposition of two group elements, one belonging to $\mathcal{F}$ and the other to $\mathcal{G}/\mathcal{F}$, i.e., $\hat{\mathbf{g}} = \hat{\mathbf{\Omega}}\hat{\mathbf{f}}$ with $\hat{\mathbf{g}} \in \mathcal{G}$, $\hat{\mathbf{f}} \in \mathcal{F}$, $\hat{\mathbf{\Omega}} \in \mathcal{G}/\mathcal{F}$;

(o2) the GCS

$$|\mathbf{\Omega}\rangle := \hat{\mathbf{\Omega}}|G\rangle \ , \ \forall\hat{\mathbf{\Omega}} \in \mathcal{G}/\mathcal{F} ; \tag{37}$$

(o3) a measure $d\mu(\hat{\mathbf{\Omega}})$ on $\mathcal{G}/\mathcal{F}$ that is invariant under the action of the elements of $\mathcal{G}$, and therefore dubbed invariant measure, such that a resolution of the identity upon $\mathcal{H}$ is provided

$$\int_{\mathcal{G}/\mathcal{F}} d\mu(\hat{\mathbf{\Omega}})|\mathbf{\Omega}\rangle\langle\mathbf{\Omega}| = \hat{\mathbb{I}}_{\mathcal{H}} \ . \tag{38}$$

The GCS are normalized, $\langle\mathbf{\Omega}|\mathbf{\Omega}\rangle = \langle G|\hat{\mathbf{g}}^{-1}\hat{\mathbf{g}}|G\rangle = \langle G|G\rangle = 1$, $\forall \ \hat{\mathbf{g}} \in \mathcal{G}$, but non-orthogonal,

$$\langle\mathbf{\Omega}|\mathbf{\Omega}'\rangle = \langle G|\hat{\mathbf{\Omega}}^{-1}\hat{\mathbf{\Omega}}'|G\rangle = \langle G|\hat{\mathbf{g}}^{-1}\hat{\mathbf{g}}'|G\rangle e^{i\theta} = \langle G|\hat{\mathbf{g}}''|G\rangle e^{i\theta} \neq 0,$$

$\forall \ \hat{\mathbf{g}}, \hat{\mathbf{g}}', \hat{\mathbf{g}}'' \in \mathcal{G}$, and $\hat{\mathbf{\Omega}}, \hat{\mathbf{\Omega}}' \in \mathcal{G}/\mathcal{F}$. For this reason, they are said to provide an "overcomplete" set of states for $\mathcal{H}$, where "complete" refers to Eq. (38), while "over" means that they are too many for being all orthogonal to each other.

As for the reference state $|G\rangle$, a common, yet not mandatory, choice is that of taking it as an extremal state; for instance, one can choose $|G\rangle$ as the minimal-weight state such that $\hat{R}_m|G\rangle = 0 \ \forall m$, with $\hat{R}_m$ defined as below.

Getting an explicit expression for the operators $\hat{\mathbf{\Omega}}$, and hence the GCS via Eq. (37), requires a characterization of the algebra. In particular, if $\mathfrak{g}$ is semisimple, one can consider its Cartan decomposition, which classifies the generators as diagonal, $\{\hat{D}_\delta\}$, or raising, $\{\hat{R}_m , \hat{R}_{-m}\}$, operators, according to

$$[\hat{D}_\delta, \hat{D}_\theta] = 0 \ , \ [\hat{D}_\delta, \hat{R}_m] = d_{\delta m}\hat{R}_m \ ,$$
$$[\hat{R}_m, \hat{R}_{-m}] = \sum_\delta d_{\delta m}\hat{D}_\delta \ , \ [\hat{R}_m, \hat{R}_{m'}] = c_{mm'}\hat{R}_{m+m'} \ . \tag{39}$$

where $\{d_{\delta m}\}$, $\{c_{mm'}\}$ are the so called structure constants, while $m, m'$ and $\delta, \theta$ go from 1 to some upper value $M$ and $D$, respectively, that depend on the algebra itself (in the case of $\mathfrak{su}(2)$, for instance, it is $M = D = 1$, and if spin-squeezing is also considered, it is $M = 2$ and $D = 1$). In any irreducible representation of $\mathfrak{g}$, it is possible to choose the raising operators such that $\hat{R}_m^\dagger = \hat{R}_{-m} \forall m$, and, consistently, Hermitian or anti-Hermitian diagonal operators $\hat{D}_\delta^\dagger = +(-)\hat{D}_\delta \ \forall \ \delta$, depending on the structure constants $\{d_{\delta m}\}$ being real or imaginary. The diagonal operators have the reference state among their eigenstates, i.e., $\hat{D}_\delta|G\rangle = g_\delta|G\rangle \ \forall \ \delta$. Once the Cartan decomposition is available, it can be shown that the elements of $\mathcal{G}/\mathcal{F}$ in the

definition (37) take the form

$$\hat{\mathbf{\Omega}} = \exp\left(\sum_m \Omega_m\hat{R}_m^\dagger - \Omega_m^*\hat{R}_m\right) , \tag{40}$$

where the coefficients $\Omega_m \in \mathbb{C}$ are coordinates of one point $\mathbf{\Omega}$ of the differentiable manifold $\mathcal{M}$, which is associated with $\mathcal{G}/\mathcal{F}$ via the quotient manifold theorem[40]. Using a complex projective representation of $\mathcal{G}/\mathcal{F}$, GCS can also be written as

$$|\mathbf{\Omega}\rangle = N(|\beta(\mathbf{\Omega})|) \ e^{\sum_m \eta_m\hat{R}_m^\dagger}|G\rangle \tag{41}$$

where the normalization constant $N(|\boldsymbol{\eta}(\mathbf{\Omega})|)$ and the relation between the $\eta_m$-coordinates and the $\Omega_m$ ones can be obtained via the BCH formulas.

The chain of biunivocal relations

$$\hat{\mathbf{\Omega}} \in \mathcal{G}/\mathcal{F} \Longleftrightarrow \mathbf{\Omega} \in \mathcal{M} \Longleftrightarrow |\mathbf{\Omega}\rangle \in \mathcal{H}. \tag{42}$$

is one of the most distinctive feature of the group-theoretic construction, as it establishes that any GCS is univocally associated with a point on $\mathcal{M}$, and vice versa. As a consequence, the invariant measure $d\mu(\hat{\mathbf{\Omega}})$ induces a measure $d\mu(\mathbf{\Omega})$ upon $\mathcal{M}$. In fact, it can be demonstrated[25] that $\mathcal{M}$ is endowed with a natural metric that can be expressed in the $\eta_m$-coordinates as

$$ds^2 = \sum_{mm'} g_{mm'} \ d\eta_m \ d\eta_{m'}^* \quad \text{where} \quad g_{mm'} := \frac{\partial^2 \log \langle\tilde{\mathbf{\Omega}}|\tilde{\mathbf{\Omega}}\rangle}{\partial\eta_m \ \partial\eta_{m'}^*} , \tag{43}$$

with $|\tilde{\mathbf{\Omega}}\rangle := |\mathbf{\Omega}\rangle /N$ in (41). After $ds^2$, one can define a canonical volume form on $\mathcal{M}$, i.e., the above-mentioned measure on $\mathcal{M}$, via

$$d\mu(\mathbf{\Omega}) = \text{const} \times \det(g)\prod_m d\eta_m \ d\eta_m^* . \tag{44}$$

The manifold $\mathcal{M}$ is also equipped with a symplectic structure that allows one to identify it as a phase-space. In particular, the symplectic form on $\mathcal{M}$ has the coordinate representation

$$\omega = -i \sum_{mm'} g_{mm'} \ d\eta_m \wedge d\eta_{m'}^* , \tag{45}$$

that can be used to define the Poisson brackets

$$\{f, g\}_{PB} := i \sum_{mm'} g^{mm'}\left(\frac{\partial f}{\partial\eta_m}\frac{\partial g}{\partial\eta_{m'}^*} - \frac{\partial f}{\partial\eta_{m'}^*}\frac{\partial g}{\partial\eta_m}\right), \tag{46}$$

with $\sum_n g_{mn}g^{nm'} = \delta_m^{m'}$.

In the case of non-semisimple algebras, such as $\mathfrak{h}_4$ and $\mathfrak{h}_6$ for the harmonic and squeezed-harmonic oscillator, respectively, where a Cartan decomposition (39) is not available, analogous decompositions exist, and the same procedure can be adopted. This is explicitly done for $\mathfrak{h}_4$ at the end of this material, where we show that the results are the same as those obtained in the semisimple case.

**Parametric representation with GCS.** Parametric representations of composite systems can be built whenever a resolution of the identity upon the Hilbert space of one of the subsystems is available. In ref. [4], for instance, the representation is introduced via $\int dx|x\rangle\langle x| = \hat{\mathbb{I}}_C$, where $|x\rangle$ are the eigenstates of the position operator for one of two subsystems, and the integral is over the real axes. Our choice, which is pivotal to get to our final result, is based on the fact that parametric representations with GCS inherit from the group-theoretic construction some properties that are essential in order to follow the quantum-to-classical crossover and formally define a classical limit of a quantum theory, according to the large-$N$ quantum approach.

The representation is defined as follows. Consider an isolated bipartite system $\Psi = C + \Gamma$ with Hilbert space $\mathcal{H}_\Psi = \mathcal{H}_C \otimes \mathcal{H}_\Gamma$, where $\Gamma$ is the principal system and $C$ its environment. The most general expression for a pure state of $\Psi$ is

$$|\Psi\rangle\rangle = \sum_{\gamma\xi} c_{\gamma\xi}|\gamma\rangle \otimes |\xi\rangle \quad \text{with} \quad \sum_{\gamma\xi} |c_{\gamma\xi}|^2 = 1 , \tag{47}$$

where $\{|\gamma\rangle\}_\Gamma$ and $\{|\xi\rangle\}_C$ are orthonormal bases for $\mathcal{H}_\Gamma$ and $\mathcal{H}_C$ respectively. Inserting the above-mentioned resolution of the identity upon $\mathcal{H}_C$, for which we choose the one provided by GCS, Eq. (38), one gets

$$|\Psi\rangle\rangle = \int_{\mathcal{M}} d\mu(\mathbf{\Omega}) \ \chi(\mathbf{\Omega}) \ |\mathbf{\Omega}\rangle \otimes |\phi(\mathbf{\Omega})\rangle , \tag{48}$$

where $\chi(\mathbf{\Omega})$ is a function that can be chosen real, being defined via $\chi^2(\mathbf{\Omega}) := \sum_\gamma |\sum_\xi c_{\gamma\xi}\langle\mathbf{\Omega}|\xi\rangle|^2$. The element $|\phi(\mathbf{\Omega})\rangle$ of $\mathcal{H}_\Gamma$ is normalized, and hence describe a pure state of $\Gamma$. Due to the normalization of $|\Psi\rangle\rangle$, it is

$$\int_{\mathcal{M}} d\mu(\mathbf{\Omega})\chi^2(\mathbf{\Omega}) = 1 , \tag{49}$$

meaning that $\chi^2(\mathbf{\Omega})$ can be interpreted as a probability distribution on $\mathcal{M}$. The above expressions have a clear physical interpretation: reminding that each point $\mathbf{\Omega} \in \mathcal{M}$ is in one-to-one correspondence with a GCS $|\mathbf{\Omega}\rangle \in \mathcal{H}_C$, we can say that $|\phi(\mathbf{\Omega})\rangle$ is the state of $\Gamma$ conditioned to $C$ being in the GCS $|\mathbf{\Omega}\rangle$, a circumstance that occurs with probability $\chi^2(\mathbf{\Omega})$ when $\Psi$ is in the pure state $|\Psi\rangle\rangle$. This interpretation

is consistent with the following relations[41]

$$\chi^2(\boldsymbol{\Omega}) = \langle \boldsymbol{\Omega} | \rho_C | \boldsymbol{\Omega} \rangle, \tag{50}$$

and

$$\rho_\Gamma = \int_{\mathcal{M}} d\mu(\boldsymbol{\Omega}) \chi(\boldsymbol{\Omega})^2 |\phi(\boldsymbol{\Omega})\rangle \ \langle \phi(\boldsymbol{\Omega})|, \tag{51}$$

where $\rho_{\Gamma(C)} := \mathrm{Tr}_{C(\Gamma)} |\Psi\rangle\rangle \ \langle\langle\Psi|$. Notice that the diagonal-like form (51) of $\rho_\Gamma$ is not generally granted for parametric representations such that the identity resolution is in terms of non-orthogonal states, as in the GCS case. In fact, it is the specific overcompleteness of GCS that ensures Eq. (51) to hold.

Finally, it is important to remind that, despite parametric representations allow one to use pure states $|\phi(\boldsymbol{\Omega})\rangle$ to describe $\Gamma$, this should by no means be intended as if $\Gamma$ were a pure state. In fact, due to the parametric dependence of $|\phi(\boldsymbol{\Omega})\rangle$ on $\boldsymbol{\Omega}$, the density operator $\rho_\Gamma$ in Eq. (51) is not a projector, reflecting that $C$ and $\Gamma$ are entangled, as far as the form (47) of $|\Psi\rangle\rangle$ stays general. To this respect, it is easily verified that when $|\Psi\rangle\rangle$ is separable the above parametric dependence dies out.

**Derivation of the symbol of $\hat{H}_C$.** In this part, we express the symbol $\langle\lambda|\hat{H}_C|\lambda\rangle$ of the clock Hamiltonian, in terms of the complex parameter $\lambda := \Omega_\ell = \varrho e^{-i\varphi}$ that defines the GCS $|\lambda\rangle$ via $|\lambda\rangle := e^{\hat{W}}|G\rangle$ with $|G\rangle$ the clock reference state satisfying $\hat{R}_m|G\rangle = 0 \, \forall m, \hat{D}_\delta|G\rangle = g_\delta|G\rangle$, and $\hat{W} := \hat{W}_\ell = \Omega_\ell \hat{R}_\ell^\dagger - \Omega_\ell^* \hat{R}_\ell$. Recalling that $\hat{H}_C = \varsigma \hat{D}_1 + k \hat{\mathbb{1}}_C$ with $k := -\varsigma g_1$ and $\varsigma^2 = \pm 1$ such that $\epsilon := \varsigma d_{1\ell}$ is real and positive, we write

$$\begin{aligned}
\langle\lambda|\hat{H}_C|\lambda\rangle &= k + \varsigma\langle G|e^{-\hat{W}}\hat{D}_1 e^{\hat{W}}|G\rangle \\
&= k + \varsigma\langle G|\hat{D}_1 + [\hat{W}, \hat{D}_1] + \frac{1}{2!}[\hat{W}, [\hat{W}, \hat{D}_1]] \\
&\quad + \frac{1}{3!}[\hat{W}, [\hat{W}, [\hat{W}, \hat{D}_1]]] \\
&\quad + \frac{1}{4!}[\hat{W}, [\hat{W}, [\hat{W}, [\hat{W}, \hat{D}_1]]]] + ...|G\rangle \\
&= k + \varsigma\langle G|\hat{D}_1 + [\hat{W}, \hat{D}_1] + \sum_\delta \Big[ \frac{1}{2!}(-2d_{1l}d_{\delta\ell}\varrho^2 \hat{D}_\delta) \\
&\quad + \frac{1}{3!}(-2d_{1\ell}d_{\delta\ell}\varrho^2[\hat{W}, \hat{D}_\delta]) \\
&\quad + \frac{1}{4!}\sum_\theta(-2d_{1\ell}d_{\theta\ell}\varrho^2)(-2d_{\theta\ell}d_{\delta\ell}\varrho^2\hat{D}_\delta) + ...]|G\rangle \\
&= \varsigma\sum_\delta g_\delta \Big[ \frac{1}{2!}(-2\varrho^2 d_{1\ell}d_{\delta\ell}) \\
&\quad + \frac{1}{4!}\sum_\theta(-2\varrho^2 d_{1\ell}d_{\theta\ell})(-2\varrho^2 d_{\theta\ell}d_{\delta\ell}) + ...] \\
&= \varsigma d_{1\ell}\sum_\delta g_\delta d_{\delta\ell} \left( \sum_{n=1}^\infty \frac{(-1)^n}{(2n)!}(\sqrt{2}\varrho)^{2n}a^{2n-2} \right) \\
&= \epsilon a^{-2}\varsigma^2 b^2 \left( \cos\left(a\sqrt{2}\varrho\right) - 1 \right) := H_C(\varrho),
\end{aligned} \tag{52}$$

where $a^2 = \sum_\theta d_{\theta\ell}^2$ and $\varsigma^2 b^2 = \sum_\delta g_\delta d_{\delta\ell}$. For the sake of a lighter notation, in what follows and in the main work we set $\varsigma^2 a^2 = 2$, which means that the raising and diagonal operators are multiplied by $\sqrt{2\varsigma^2}/a$, and their eigenvectors are rescaled accordingly. We thus finally get

$$\langle\lambda|\hat{H}_C|\lambda\rangle = \frac{\epsilon b^2}{2}(\cos(2\varsigma\varrho) - 1). \tag{53}$$

**The pullback-by-$F$ and the Poisson brackets on $\mathcal{M}_C$.** In this part, we will explicitly calculate the Poisson brackets $\{\cdot, \cdot\}_C$ induced on $\mathcal{M}_C$ via the pullback-by-$F$. We recall that, given the manifolds $\mathcal{M}_C$ and $\mathcal{M}_\Gamma$ for the clock $C$ and the evolving system $\Gamma$ as from the GCS construction, the map $F: U_C \subset \mathcal{M}_C \to U_\Gamma \subset \mathcal{M}_\Gamma$, is defined as

$$\begin{cases} q_j = \sqrt{2} \, b \, \varsigma \, \sin(\varsigma\varrho) \cos(\varphi) \, v_j, \\ p_j = \frac{\sqrt{2} \, b}{\varsigma} \, \sin(\varsigma\varrho) \sin(\varphi) \, v_j, \end{cases} \tag{54}$$

with $\sum_j v_j^2 = 1$. We remind that the Poisson brackets are defined on a generic symplectic manifold $\mathcal{M}$ starting from its symplectic form $\omega = \frac{1}{2}\sum_{\mu\nu}\omega_{\mu\nu}dx^\mu \wedge dx^\nu$, via $\{f, g\} = \sum_{\mu\nu}\omega^{\mu\nu} \, \partial_{x_\mu}f \, \partial_{x_\nu}g$ with $\sum_\sigma \omega_{\mu\sigma} \, \omega^{\sigma\nu} = \delta_\mu^\nu$, and $x^\mu(\mu = 1, ..., 2n = \dim\mathcal{M})$. $f, g$ are some coordinates and generic functions on $\mathcal{M}$, respectively. In fact, the Darboux theorem guarantees that there exists local coordinates $x^\mu = (q_1, ..., q_n, p_1, ..., p_n)$ such that $\omega = \mathfrak{h} \sum_{j=1}^n dp_j \wedge dq_j$ and $\{f, g\} = \mathfrak{h}^{-1} \sum_{j=1}^n \left( \partial_{q_j}f \, \partial_{p_j}g - \partial_{p_j}f \, \partial_{q_j}g \right)$, with $\mathfrak{h} = const.$. This said, being $(q_j, p_j)$ in Eq. (54) Darboux coordinates, i.e.,

$\{q_j, p_j\}_\Gamma = \mathfrak{h}^{-1}\delta_{ij}$, the symplectic form $\omega_\Gamma$ on $U_\Gamma \subset \mathcal{M}_\Gamma$ is

$$\omega_\Gamma = \mathfrak{h} \sum_j dp_j \wedge dq_j. \tag{55}$$

We can now calculate the pullback-by-$F$ of $\omega_\Gamma$ as

$$\begin{aligned}
(\omega_\Gamma)^* &= \mathfrak{h} \sum_j \Big[ \sqrt{2}b \, \cos(\varsigma\varrho)\sin(\varphi) \, v_j \, d\varrho \\
&\quad + \frac{\sqrt{2}b}{\varsigma} \sin(\varsigma\varrho)\cos(\varphi) \, v_j \, d\varphi \Big] \\
&\quad \wedge \Big[ \sqrt{2} \, b\varsigma^2 \, \cos(\varsigma\varrho)\cos(\varphi) \, v_j \, d\varrho \\
&\quad - \sqrt{2}b\varsigma \, \sin(\varsigma\varrho)\sin(\varphi) \, v_j \, d\varphi \Big] \\
&= \mathfrak{h} \sum_j \Big[ -b^2 \, \varsigma \, \sin(2\varsigma\varrho) \, v_j^2\sin^2\varphi \, d\varrho \wedge d\varphi \\
&\quad + b^2 \, \varsigma \, \sin(2\varsigma\varrho) \, v_j^2 \, \cos^2\varphi \, d\varphi \wedge d\varrho \Big] \\
&= \mathfrak{h}b^2\varsigma \, \sin(2\varsigma\varrho) \, d\varphi \wedge d\varrho.
\end{aligned} \tag{56}$$

Finally $(\omega_\Gamma)^*$ defines Poisson brackets on $\mathcal{M}_C$ via

$$\{f_C, g_C\}_C = \frac{1}{\mathfrak{h}b^2 \, \varsigma \, \sin(2\varsigma\varrho)} \left( \frac{\partial f_C}{\partial\varrho}\frac{\partial g_C}{\partial\varphi} - \frac{\partial f_C}{\partial\varphi}\frac{\partial g_C}{\partial\varrho} \right). \tag{57}$$

We clarify that our choice (54) for the map $F$ follows from the one suggested in ref. [25], but other choices are possible.

**The Heisenberg algebra $\mathfrak{h}_4$.** When the Lie algebra $\mathfrak{g}$, to which the clock Hamiltonian $\hat{H}_C$ belongs, is semisimple, the GCS are built starting from the Cartan decomposition. However, a similar construction can be put forward for the non-semisimple algebra $\mathfrak{h}_4$. The latter is defined by the set $\{\hat{n} = \hat{a}^\dagger\hat{a}, \hat{a}, \hat{a}^\dagger, \hat{\mathbb{1}}\}$ with commutation relations $[\hat{a}, \hat{a}^\dagger] = \hat{\mathbb{1}}, \quad [\hat{a}, \hat{\mathbb{1}}] = [\hat{a}^\dagger, \hat{\mathbb{1}}] = 0$. The GCS $|\alpha\rangle$, usually called harmonic-oscillator coherent states or just coherent states, are in one-to-one correspondence with the points of the complex plane $\mathbb{C}$ and can be equivalently defined as $|\alpha\rangle = e^{\alpha\hat{a}^\dagger - \alpha^*\hat{a}}|G\rangle = e^{-|\alpha|^2/2} \, e^{\alpha\hat{a}^\dagger}|G\rangle$ with $\hat{n}|G\rangle = \hat{a}|G\rangle = 0$, or as $\hat{a}|\alpha\rangle = \alpha|\alpha\rangle$ with $\alpha \in \mathbb{C}$. When the clock $C$ admits a proper classical limit, the Schrödinger equation for the evolving system $\Gamma$ can be obtained, as shown in the main work, implementing the PaW mechanism via the parametric representation with GCS, and considering a fixed GCS $|\lambda\rangle = e^{\lambda\hat{a}^\dagger - \lambda^*\hat{a}}|G\rangle = N_\varrho e^{\lambda\hat{a}^\dagger}|G\rangle$ with $\lambda = \varrho e^{i\varphi}$, for which, being $[\hat{n}, e^{\lambda^*\hat{a}}] = -\lambda^*\hat{a}e^{\lambda^*\hat{a}}$, it is $\langle\lambda|\hat{H}_C|\alpha\rangle = i\epsilon\frac{d}{d\varphi}\langle\lambda|\alpha\rangle$, where $\hat{H}_C = \epsilon\hat{n}$. Again, the temporal parameter $t^{QM}$ turns out to be $t^{QM} = (\hbar/\epsilon)\varphi$. Moreover, since it is trivial to show that $H_C(\lambda) = \langle\lambda|\hat{H}_C|\lambda\rangle = \epsilon\varrho^2$, the considerations concerning the parameter $\varrho$ and the stationary Schrödinger equation for $\Gamma$ still apply. For what concerns the uncertainty relation, a phase-operator can be defined via $\hat{a} = \hat{n}^{1/2}e^{i\hat{\phi}}$. Finally, when $\Gamma$ becomes macroscopic and presents a completely classical behavior, its dynamics is ruled by the Hamilton equations according to a temporal parameter $t^{CL} = t^{QM}$. This result can be obtained following the same line of reasoning of the main work and choosing the map $F$ to be $q_j - ip_j = v_j\sqrt{2}\varrho e^{i\varphi}$ with $\sum_j v_j^2 = 1$.

## Data availability
No datasets were generated or analyzed during the current study.

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

## Acknowledgements

We thank G. García-Pérez, S .Maniscalco, B. Sokolov, and F. Bonechi for amazing discussions, stimulating questions, and valuable suggestions. C.F. and P.V. also acknowledge the beauty of the Dolomites for inspiration during the Quantum Hiking 2019 international conference. P.V. gratefully acknowledges support from the Kavli Institute for Theoretical Physics @UC Santa Barbara, in the framework of the program Open Quantum System Dynamics, and all the discussions with the participants during the program itself. A. Cuccoli and C.F. acknowledge Fondazione CR Firenze for financial support within project 2018.0951. This work is done in the framework of the Convenzione operativa between the Institute for Complex Systems of the Consiglio Nazionale delle Ricerche (Italy) and the Physics and Astronomy Department of the University of Florence.

## Author contributions

All authors contributed to all aspects of the research.

## Competing interests

The author declares no competing interests.
