## [Peer Review File · Nature Communications]

Reviewers' Comments:

Reviewer #1:

Remarks to the Author:

The Authors extend the Page and Wootters construction of "time without time" by using coherent states and the large N limit. With respect to the original proposal, in the current extension there is no need to introduce time measurements to make the time a classical variable, as one can perform the classical limit through excited coherent states. Also, a nice derivation of an interesting new type of time-energy uncertainty relation is given.

I think the idea is really nice and is potentially a large contribution in the understanding of the problem of time in quantum mechanics. I suggest the paper be published in Nature Communications.

There are a few points that perhaps could be clarified:

1. I do not see any connection to general relativity or gravity that is promised in the abstract, except for a couple of very speculative sentences in the conclusions. I believe that the abstract should not contain any reference to general relativity or gravity, as there are no results to this effect in the current paper.
2. I am confused on the role of the function $\chi(\Omega)$ in Eq.(10). The Authors seem to put no requirements on it (except for the fact that it is real and its square is normalized), but that cannot work. For example, if one chooses a delta function (or a very narrow Gaussian) the entanglement in the state (3) is lost. Then we cannot move the dependence from ϕ (the time parameter) to the state $|\phi(\Omega)\rangle$ in Eq.(10). Indeed, the derivation above Eq.(10) suggests that $\chi(\Omega)$ should not be null for any values of the chosen Ω_I . Also the normalization condition given in (17) suggests the same. Perhaps this should be clarified and it should be explained what is the role of χ in the construction.
3. If I understand correctly, the choice of I in Eq.(5) determines which, among the M -tuples is the clock (the other ones refer to the other degrees of freedom in the symplectic manifold). The Authors just simply state that I can be "chosen at will". Is the choice of which variable to choose as time completely arbitrary as the Authors seem to imply? Perhaps a comment on this would be useful.
4. Below Eq.(6) a $\tan(\zeta \rho)$ can only refer to $\mathfrak{su}(2)$: that normalization factor comes from the BCH and only the $\mathfrak{su}(2)$ BCH contains a tangent. So that should be removed, unless the Authors want to restrict the discussion to $\mathfrak{su}(2)$ (I doubt that).
5. The Authors refer to $\Delta \phi$ "as a time-interval". As far as I can see, this is not the case: from their derivation, i.e. from Eq.(14), it seems that $\Delta \phi$ is a standard deviation. Referring to $\Delta \phi$ as a time interval is quite confusing because it indeed is a time interval in the customary time-energy uncertainty relations. However, here it is not, if I understand correctly. This should be clarified. Also, optionally, it might be nice if it was

possible to relate this uncertainty relation with the customary time-energy uncertainty relation, as this one seems to have a completely different derivation and scope. Perhaps this might be too far from the scope of the paper, so I am not suggesting it as a strict requirement.

In conclusion, I suggest the paper to be published in Nature Communications once the above minor issues are addressed, which shouldn't be too complicated.

Reviewer #2:

Remarks to the Author:

The authors study "the problem of quantum time", that is, how time emerges from a stationary quantum state. They showed under some assumptions on a clock and a system a time parameter is defined, with respect to which the Schroedinger equation and the classical Hamilton equation are derived.

While analysis in the paper is mathematically sound as far as I read, there remain several problems for it to be published as it is.

The authors employ the Page-Wootters formalism naively without taking into account several critics such as "wrong propagator issue" against it. As far as I understand, the present paper is not trying to tackle them, thus it may not be attracting many audiences.

The exposition is a bit too mathematical. Not so many readers know about Cartan decomposition while everyone knows ladder operators in $SU(2)$ irreducible representation and creation annihilation operators of a harmonic oscillator. Considering the broad audience of this journal and importance of examples in physics, I recommend that the authors should start with concrete examples of coherent states and end with its general formalism.

As a whole, if we accept the assumptions such as PaW formalism itself, existence of somehow robust entangled state, somehow a clock system possessing a group large enough to allow coherent states, etc., we are led to their conclusions.

In fact their conclusions are natural because a coherent state in the large N limit well describes quantum-classical correspondence. But I could not find any strong reason to believe the assumptions themselves.

”There is only one time”: Replies to the Referees’ questions, criticisms, and comments

Caterina Foti, Alessandro Coppo, Giulio Barni, Alessandro Cuccoli, and Paola Verrucchi

(Dated: November 6, 2020)

FIRST REFEREE

We thank the referee for expressing approval for our work and providing valuable comments on the manuscript, that have helped us making it clearer and more precise. A detailed analysis of the Referee’s Report is given below, together with our replies and the changes made to the manuscript, accordingly.

1.

The referee writes:

I do not see any connection to general relativity or gravity that is promised in the abstract, except for a couple of very speculative sentences in the conclusions. I believe that the abstract should not contain any reference to general relativity or gravity, as there are no results to this effect in the current paper.

Our reply:

The referee is right.

Changes made:

The sentence at issue has been removed from the abstract.

2.

The referee writes:

I am confused on the role of the function $\chi(\Omega)$ in Eq.(10). The Authors seem to put no requirements on it (except for the fact that it is real and its square is normalized), but that cannot work. For example, if one chooses a delta function (or a very narrow Gaussian) the entanglement in the state (3) is lost. Then we cannot move the dependence from ϕ (the time parameter) to the state $|\phi(\Omega)\rangle$ in Eq.(10). Indeed, the derivation above Eq.(10) suggests that $\chi(\Omega)$ should not be null for any values of the chosen Ω . Also the normalization condition given in (17) suggests the same. Perhaps this should be clarified and it should be explained what is the role of χ in the construction.

Our reply:

We thank the referee for raising this point, as it makes us understand that the definition of $\chi(\Omega)$ is not clear in the original manuscript. In fact, the distribution $\chi(\Omega)$ is not chosen, neither arbitrarily nor independently: its specific form is set by the state of the overall system $|\Psi\rangle$. This follows from the definition $\chi^2(\Omega) = \sum_{\gamma} |\sum_{\xi} c_{\gamma\xi} \langle \Omega | \xi \rangle|^2$, given in the paragraph below Eq.(48) in the original manuscript. Here, the sum over the index γ shows that as far as the state is entangled, implying that more than one coefficient $c_{\gamma\xi}$ is different from zero, $\chi^2(\Omega)$ cannot be, or become in some limit, a unimodal distribution. Referring to the example suggested by the referee, consider the case when the states $|\xi^{\gamma}\rangle$ are GCS themselves, labeled by γ , and $c_{\gamma\xi}$ thus only depends on γ (as it happens in the dynamics of the quantum-measurement process described by the Ozawa-Von Neumann model for ideal PVM): for $N \rightarrow \infty$, each overlap $\langle \Omega | \xi^{\gamma} \rangle$ is indeed bound to become proportional to a δ -function, but each overlap entering the definition of χ^2 is weighted by $|c_{\gamma}|^2$, which ensures the normalization of χ^2 and passes on the entangled nature of the original state $|\Psi\rangle$ to the large- N limit. To this respect, several examples are shown in Refs.[24],[32], and [34]. Summarising: the Referee is right in saying that if $\chi(\Omega)$ is a delta-function the entanglement is lost and the system cannot work as a clock: however, if there is entanglement and C can hence work as a clock, $\chi(\Omega)$ cannot become a single delta-function, but it rather tends, as its definition implies, to a superposition of different unimodal distributions. In order to make all of the above clearer in the manuscript, we have modified the second paragraph of the Section RESULTS, so as to include both an explicit definition of the overall state $|\Psi\rangle$ and that of $\chi(\Omega)^2$.

Changes made:

The paragraph containing Eq. (2) is split; two sentences are added after the one containing Eq. (2); one sentence is added in the paragraph containing Eq.(3). In particular

- @line 105

”... a lighter notation. In view of...”

is now

”... a lighter notation. As for the state $|\Psi\rangle\rangle$, its most general expression is $|\Psi\rangle\rangle = \sum_{\gamma\xi} |\xi\rangle \otimes |\gamma\rangle$, where $\{|\xi\rangle\}_C$ and $\{|\gamma\rangle\}_\Gamma$ are orthonormal bases of \mathcal{H}_C and \mathcal{H}_Γ , respectively; the coefficients $c_{\gamma\xi} \in \mathbb{C}$ are such that $\sum_{\gamma\xi} |c_{\gamma\xi}|^2 = 1$ due to the normalization of $|\Psi\rangle\rangle$. Notice that if $|\Psi\rangle\rangle$ is entangled, there cannot exist orthonormal bases $\{|\xi\rangle\}_C$ and $\{|\gamma\rangle\}_\Gamma$ such that only one coefficient $c_{\gamma\xi}$ is different from zero.

In view of ...”

- @line 118 to 123

”... resolution of the Identity. The positive function... is entangled. There is a certain degree of freedom...”

is now

”... resolution of the Identity. The element $|\phi(\Omega)\rangle \in \mathcal{H}_\Gamma$ is normalized, and hence describes a physical state of Γ , parametrically dependent on Ω . Notice that the Ω -dependence of $|\phi(\Omega)\rangle$ survives iff $|\Psi\rangle\rangle$ is entangled. As for $\chi(\Omega)$, it is defined [25] (up to an arbitrary phase-factor) through

$$\chi^2(\Omega) = \sum_{\gamma} \left| \sum_{\xi} c_{\gamma\xi} \langle \Omega | \xi \rangle \right|^2,$$

and can hence be taken real without loss of generality. The above function $\chi^2(\Omega)$ is a normalized probability distribution on \mathcal{M}_C whose structure is strongly related to the entanglement property of $|\Psi\rangle\rangle$; in particular, if $|\Psi\rangle\rangle$ is entangled, $\chi^2(\Omega)$ is a superposition of different unnormalized distributions $|\sum_{\xi} c_{\gamma\xi} \langle \Omega | \xi \rangle|^2$.

There is a certain degree of freedom ...”

3.

The Referee writes:

If I understand correctly, the choice of l in Eq.(5) determines which, among the M -tuples is the clock (the other ones refer to the other degrees of freedom in the symplectic manifold). The Authors just simply state that l can be ”chosen at will”. Is the choice of which variable to choose as time completely arbitrary as the Authors seem to imply? Perhaps a comment on this would be useful.

Our reply:

We thank the referee for this comment. In fact, the choice of ℓ determines which observable/degree-of-freedom of the clock carries information about the coordinate that plays the role of time, and how. Formally, ℓ can be chosen at will as long as it ensures ϵ real and positive, and $\chi(\Omega_\ell) \neq 0$. On the other hand, the freedom left by this mild request allows one to further ask that ”time” could be read by observing the clock. This means that there must exist hermitian linear combinations of the operators \hat{R}_ℓ and \hat{R}_ℓ^\dagger that are experimentally accessible observables for C such that the results of their measurements carry information upon the coordinate φ . In order to better clarify this point, we have slightly modified the sentence above Eq.(5) and added a sentence at the end of the second-to-last paragraph of the section *A quantum clock for a quantum system*

Changes made:

- @line 165

” ... with ℓ chosen at will amongst those for which ϵ is real and positive.”

is now

” ... with ℓ chosen at will amongst those for which ϵ is real and positive and $\chi(\Omega_\ell) \neq 0$.”

- @line 207

”... that is made clear in the next section.”

is now

”... that is made clear in the next section. We also remind that the choice of ℓ is arbitrary, as long as it ensures ϵ real and positive, and $\chi(\Omega_\ell) \neq 0$. So, the freedom left by this mild request allows one to further ask that time can be read by observing the clock. This means that there must exist hermitian linear combinations of the operators \hat{R}_ℓ and \hat{R}_ℓ^\dagger that are experimentally accessible observables for C , and such that the results of their measurements carry information upon the coordinate φ .”

4.

The referee writes:

4. Below Eq.(6) a tan(zeta rho) can only refer to su(2): that normalization factor comes from the BCH and only the su(2) BCH contains a tangent. So that should be removed, unless the Authors want to restrict the discussion to su(2) (I doubt that).

Our reply:

Indeed we do not implement one such restriction, The key point to address this comment is the definition of the parameter ς , that follows from $\varsigma^2 = \pm 1$ (see the paragraph above the one containing Eq.(4)). This definition implies either $\varsigma = \pm 1$ or $\varsigma = \pm i$, with i imaginary unit. The first equality holds when the manifold \mathcal{M}_C is compact (as in the case of $\mathfrak{su}(2)$), and the second one when it is non-compact (as in the case of $\mathfrak{su}(1, 1)$). As a consequence, depending on whether \mathcal{M}_C is compact or not, the function with argument proportional to ς are either trigonometric or hyperbolic, respectively (as shown in section 3C.1.a of Ref.[25]). In order to make this point clearer, we have added a sentence after the introduction of ς^2 , in the paragraph above the one containing Eq.(4).

Changes made:

- @line 147-148

"... is Hermitian. Notice that, due to the above definition of H_C ..."

is now

"... is Hermitian. Notice that $\varsigma^2 = +1$ or -1 depending on whether \mathcal{M}_C is compact or non-compact, respectively; consequently, ς is either 1 or i and the functions having argument proportional to ς have a different nature, trigonometric or hyperbolic, respectively, throughout the manuscript (see Sec.3C.1.a of Ref. [25] for more details). As for the parameter ϵ , due to the definition of H_C ..."

5.

The referee writes:

The Authors refer to Delta phi "as a time-interval". As far as I can see, this is not the case: from their derivation, i.e. from Eq.(14), it seems that Delta phi is a standard deviation. Referring to Delta phi as a time interval is quite confusing because it indeed is a time interval in the customary time-energy uncertainty relations. However, here it is not, if I understand correctly. This should be clarified. Also, optionally, it might be nice if it was possible to relate this uncertainty relation with the customary time-energy uncertainty relation, as this one seems to have a completely different derivation and scope. Perhaps this might be too far from the scope of the paper, so I am not suggesting it as a strict requirement.

Our reply:

We thank the referee for raising this point. Indeed $\Delta\varphi$ in the inequality (24), as well as in the rest of the manuscript, is a standard deviation, as correctly understood by the referee after its derivation from Eq.(14). Therefore, our writing "... and consequently $\Delta\varphi$ as a time-interval..." after (24) is wrong, and has been removed. In that same paragraph, we have added a brief comment about the different nature of our result. Consistently, we have added a related sentence at the end of the second paragraph of the concluding section.

Changes made:

- @line 276 to 280

"... of the clock itself. However, if ... uncertainty relation. We will further..."

is now

"... of the clock itself. If the parameter $\hbar\varphi/\epsilon$ is identified with time, the above inequality has the form of an energy-time uncertainty relation for Γ . However, $\Delta\varphi$ is not a time-interval, and the way the result (24) is obtained suggests an altogether different interpretation, w.r.t. the usual one, of the elements entering the energy-time uncertainty relation. We will further ..."

- @line 410 to 412

"... quantum feature, namely entanglement.

What is most remarkable, though, is that ..."

is now

“... quantum feature, namely entanglement. It is also worth reminding that $\Delta\varphi$ in the inequality (24) does not emerge as a time-interval, which suggests that such inequality, despite having the same form of the customary energy-time uncertainty relation, is of a somewhat different nature and origin. This relevant point will be further investigated in future work.

Overall, it is indeed remarkable that...”

SECOND REFEREE

We thank the referee for validating the mathematical soundness of the analysis presented in our manuscript. We have carefully considered the problems that the Referee indicates as obstacles to the publication of our work as it is, and acted consequently on the text. A detailed analysis of the Referee’s Report is given below, together with our replies and the changes made to the manuscript.

1.

The referee writes:

The authors employ the Page-Wootters formalism naively without taking into account several critics such as “wrong propagator issue” against it. As far as I understand, the present paper is not trying to tackle them, thus it may not be attracting many audiences.

Our reply:

As the referee correctly understands, our work is developed within the framework provided by the so called “PaW mechanism”, as introduced by D.N. Page and W.K. Wootters. We are fully aware that their proposal is not accepted with one voice, and that several criticisms have been raised against it. However, each of the assumptions underlying the mechanism, and many of the consequences implied by it, have been extensively scrutinized in the literature, as seen for instance in our Refs. [4]-[5], and thoroughly reported in Ref.[11] and in the more recent “The Trinity of Relational Quantum Dynamics”, by P.A. Hoehn, A.R.H. Smith, and M.P.E. Lock (arxiv:1912.00033). As for the the “wrong propagator” issue raised by Kuchař and mentioned by the Referee, it is indeed addressed, for instance, in our Ref.[4] as well as in the above mentioned arxiv:1912.00033.

All of the above said, aim of our work is not that of discussing the PaW mechanism, as clearly stated in the Introduction, but rather to use it, without however hiding the rich debate around it, as clearly declared by our Bibliography.

For this reason, we are quite confused by the Referee’s sentence: “...the present paper is not trying to tackle them, **THUS** it may not be attracting many audiences”, as we believe that the derivation of relevant consequences of the PaW mechanism, as those presented in our manuscript, should be of interest to an audience at least as large as that involved in the discussion of the mechanism itself.

In order to make our viewpoint about this issue as clear as possible we have added a sentence at the end of the first paragraph of the Introduction

Changes made:

- @line 47

“... the experimental viewpoint [2–4, 11–22].”

is now

” ... the experimental viewpoint[2-16]. Discussing about the many questions and answers on the PaW mechanism is not the scope of this work: however, references throughout the paper should help the reader to navigate the relevant literature, and comments about the contribution that our results furnish to the overall discussion on the mechanism itself are reported in the concluding section.”

2.

The referee writes:

The exposition is a bit too mathematical. Not so many readers know about Cartan decomposition while everyone knows ladder operators in $SU(2)$ irreducible representation and creation annihilation operators of a harmonic oscillator. Considering the broad audience of this journal and importance of examples in physics, I recommend that the authors should start with concrete examples of coherent states and end with its general formalism.

Our reply:

We thank the referee for this precious suggestion: We have modified the manuscript accordingly, giving examples of more familiar algebras first, and moving the introduction of the Cartan decomposition to a later paragraph.

Changes made:

- @line 122 to 142

”There is a certain degree of freedom in the group-theoretic construction of GCS... at the end of the section METHODS”
is now

”There is a certain degree of freedom in the group-theoretic construction of GCS (see for instance Tables I and II in Ref.[25]), due to the possibility of choosing an arbitrary state $|G\rangle$ from which to start the construction, the so called reference state, and different sets of generators for \mathfrak{g}_c . For the non-semisimple algebra \mathfrak{h}_4 defining the harmonic-oscillator coherent states, for instance, it is customary to choose the set $\{\hat{a}^\dagger, \hat{a}, \hat{a}^\dagger \hat{a}, \hat{\mathbb{I}}\}$. When the semisimple Lie algebras $\mathfrak{su}(2)$ or $\mathfrak{su}(1,1)$, defining the spin or pseudo-spin coherent states respectively, are considered, the standard choice is the set $\{\hat{S}^-, \hat{S}^+, \hat{S}_z\}$ in the first case and $\{\hat{K}^-, \hat{K}^+, \hat{K}_0\}$ in the second one, being $\hat{S}_z \setminus \hat{K}_0$ diagonal operators and $\hat{S}^-, \hat{S}^+ \setminus \hat{K}^-, \hat{K}^+$ raising ones. In fact, in the case of semisimple algebras different choices of the generators are generally embedded in a different choice of the so called Cartan decomposition. To this respect, we remind that the Cartan decomposition classifies generators into *diagonal*, $\{\hat{D}_\delta\}$, and *raising* operators, $\{\hat{R}_m, \hat{R}_{-m}\}$, according to $[\hat{D}_\delta, \hat{D}_\theta] = 0, [\hat{D}_\delta, \hat{R}_m] = d_{\delta m} \hat{R}_m, [\hat{R}_m, \hat{R}_{-m}] = \sum_\delta d_{\delta m} \hat{D}_\delta$, and $[\hat{R}_m, \hat{R}_{m'}] = c_{mm'} \hat{R}_{m+m'}$, where the coefficients $\{d_{\delta m}\}, \{c_{mm'}\}$ are the so called structure constants. Referring again to the familiar cases of $\mathfrak{su}(2)$ and $\mathfrak{su}(1,1)$, it is $M = 1$, with $\hat{R}_1 = \hat{S}^-$ and \hat{K}^- . If spin squeezing is also considered, it is $M = 2$, with $\hat{R}_2 = (\hat{S}^-)^2$. Let us finally remind that the harmonic-oscillator algebra \mathfrak{h}_4 is not semisimple, and a Cartan decomposition cannot hence be defined for it. Nevertheless, the usual choice of generators corresponds to an analogous decomposition, with $(\hat{a}^\dagger \hat{a}, \hat{\mathbb{I}})$ diagonal and $(\hat{a}^\dagger, \hat{a})$ creation/annihilation operators. Given the relevance of the harmonic-oscillator coherent states in the literature, we have considered their specific case in the last section of METHODS.

3.

The referee writes:

As a whole, if we accept the assumptions such as PaW formalism itself, existence of somehow robust entangled state, somehow a clock system possessing a group large enough to allow coherent states, etc., we are led to their conclusions. In fact their conclusions are natural because a coherent state in the large N limit well describes quantum-classical correspondence. But I could not find any strong reason to believe the assumptions themselves.

Our reply:

We thank the referee for writing that our conclusions are ”natural”: in fact, we believe that, no matter how complicated the mathematical structure (and work) leading to a given result, its final plainness is what can make it relevant to the large audience, and we have worked hard just to this aim.

On the other hand, we think that the judgement on our work should not be conditioned by the specific position taken in the articulated debate about the ”PaW mechanism”. Rather, the other way round: the fact that the classical eom of Hamilton can be derived within a fully quantum picture, without even knowing about classical physics, and with no assumptions other than those of the PaW mechanism, is a strong argument in favour of the latter. To make this point as clear as possible, besides the changes mentioned above in relation to the first criticism of the Referee, we have also added a comment after the first sentence of the third paragraph of the concluding section.

Changes made:

- @line 418

” ... Hamilton e.o.m (34). In fact our results...”

is now

” ... Hamilton e.o.m (34). Indeed, the fact that fundamental laws of classical Physics be derived within a completely quantum framework with the PaW-mechanism assumptions enforced, substantiates the mechanism itself. Moreover, our results...”

Reviewers' Comments:

Reviewer #1:

Remarks to the Author:

I believe that the Authors have replied satisfactorily to the issues I and the other Referee had raised. As for the other Referee's assessment, I believe that the Referee's greatest issue refers to whether the Authors address the criticisms that had been raised in the past against the Page & Wootters mechanism. As also the Authors point out, I believe that these criticisms have been more than satisfactorily addressed in other publications (some of whom are very recent, the Referee may not be aware of them) that are clearly referenced in this paper, so there is no need to address them also here.

In conclusion, I suggest that the paper be published in Nature Communications.